# IndiaWeatherBench: A Dataset and Benchmark for Regional Weather Forecasting over India

## Abstract

Regional weather forecasting is a critical problem for localized climate adaptation, disaster mitigation, and sustainable development. While machine learning has shown impressive progress in global weather forecasting, regional forecasting remains comparatively underexplored. Existing efforts often use different datasets and experimental setups, limiting fair comparison and reproducibility. We introduce IndiaWeatherBench, a comprehensive benchmark for data-driven regional weather forecasting focused on the Indian subcontinent. IndiaWeatherBench provides a curated dataset built from high-resolution regional reanalysis products, along with a suite of deterministic and probabilistic metrics to facilitate consistent training and evaluation. We establish strong baselines by adapting state-of-the-art global models, including FourCastNet, Pangu-Weather, GraphCast, and Stormer, to the regional domain. To enable this adaptation, we propose two simple yet effective boundary conditioning strategies: boundary forcing and coarse-resolution conditioning. We conducted a thorough empirical evaluation of these baselines under different settings and metrics, complemented by a case study on predicting extreme heatwaves in India. While focused on India, we designed IndiaWeatherBench to be easily extensible to other geographic regions. We will open-source all raw and preprocessed datasets, model implementations, and evaluation pipelines to promote accessibility and future development in regional weather forecasting research.

## 1 Introduction

The increasing frequency, intensity, and impact of extreme weather events such as heatwaves, floods, cyclones, and droughts underscore the urgent need for accurate and actionable weather forecasts in a changing climate. These forecasts are especially critical at the regional and local level, where governments, businesses, and communities make day-to-day decisions that depend on reliable forecasts. Traditionally, weather and climate modeling have relied on numerical methods, which simulate the evolution of the atmosphere by solving complex systems of partial differential equations over discretized spatial grids (Lynch, 2008; Bauer et al., 2015). While these numerical weather prediction (NWP) models have become indispensable tools in modern meteorology, they face persistent limitations of significant computational cost and challenges in accurately representing local geographical features and subgrid-scale processes (Stensrud, 2009).

In recent years, machine learning (ML) has emerged as a powerful alternative or complement to traditional physics-based models. Leveraging large-scale reanalysis datasets and advances in deep learning architectures, data-driven approaches have demonstrated impressive performance in various forecasting tasks – from nowcasting (Ravuri et al., 2021; Sønderby et al., 2020; Andrychowicz et al., 2023), medium-range weather forecasting (Weyn et al., 2020; Rasp & Thuerey, 2021; Keisler, 2022; Pathak et al., 2022b; Bi et al., 2022; Lam et al., 2023; Nguyen et al., 2023c; Chen et al., 2023b;a; Price et al., 2024), to climate downscaling (Baño Medina et al., 2020; Liu et al., 2020; Nagasato et al., 2021; Rodrigues et al., 2018; Sachindra et al., 2018; Vandal et al., 2019) and emulation (Kochkov et al., 2023; Watson-Parris et al., 2022; Yu et al., 2023). These models offer significantly faster inference and increasingly competitive skill scores, especially when trained on high-quality historical data. However, much of this progress has been concentrated at the global scale, largely driven by the availability of standardized, accessible benchmarks such as WeatherBench (Rasp et al., 2020), WeatherBench 2 (Rasp et al., 2023), and ChaosBench (Nathaniel et al., 2024). These benchmarks have played a pivotal role in establishing reproducible baselines, unified metrics, and community-wide leaderboards,

catalyzing rapid progress in model development. In contrast, regional weather forecasting remains comparatively underexplored in the ML community, despite its importance to real-world climate adaptation and policy planning. Moreover, regional meteorological agencies often maintain higher-quality and higher-resolution datasets than global reanalysis systems due to their focused data assimilation over limited geographic areas, which presents a promising opportunity for more accurate, fine-grained forecasting (Kaiser-Weiss et al., 2019). Yet existing regional forecasting efforts often rely on bespoke datasets, varying spatial resolutions, and inconsistent evaluation protocols (Oskarsson et al., 2023; Pathak et al., 2024; Qin et al., 2024). As a result, models are trained and tested in incompatible settings, making fair comparison difficult and limiting the development of future methods. The lack of a unified framework for regional forecasting represents a significant bottleneck to scientific progress and real-world deployment in climate-sensitive regions.

To bridge this gap, we introduce IndiaWeatherBench, a comprehensive and open benchmark for data-driven regional weather forecasting focused on the Indian subcontinent. We chose India as our region of interest not only for its immense societal relevance – home to over 1.4 billion people whose lives are closely tied to weather-sensitive sectors such as agriculture, water management, and disaster preparedness, but also for the scientific challenges it poses. The Indian region features extraordinary climatic diversity, ranging from arid deserts and high mountains to tropical rainforests and monsoon coasts, creating highly heterogeneous and dynamic weather patterns that are difficult to capture using coarse global models. To support robust model development in this complex setting, we built IndiaWeatherBench upon the IMDAA (Ashrit et al., 2020) regional reanalysis dataset that provides 12-km spatial resolution and hourly observations tailored to Indian monsoon dynamics. IndiaWeatherBench offers a preprocessed version of IMDAA with 20 years of multi-channel atmospheric states at 6-hour intervals, standardized train-validation-test splits, and a diverse suite of evaluation metrics for both deterministic and probabilistic settings. To establish strong and diverse baselines, we implement a broad range of advanced architectures, including Graphcast (Lam et al., 2023), Pangu-Weather (Bi et al., 2022), FourCastNet (Pathak et al., 2022b), and Stormer (Nguyen et al., 2023c), along with various boundary conditioning strategies and training objectives.

While geographically focused on India, IndiaWeatherBench is designed to be modular and extensible to other regions and datasets. All data preprocessing pipelines, model implementations, and evaluation code are fully open-sourced to foster transparency, reproducibility, and broad community participation. By providing the first standardized and reproducible testbed for regional ML-based weather forecasting over India, IndiaWeatherBench aims to accelerate the development of high-resolution and accurate models for high-impact, regional-scale weather prediction.

## 2  RELATED WORK

**Deep learning for weather forecasting**  Deep learning has rapidly transformed weather forecasting by providing accurate and efficient solutions across a range of prediction tasks. Models such as Pangu (Bi et al., 2022), Graphcast (Lam et al., 2023), and Stormer (Nguyen et al., 2023c) have surpassed traditional numerical systems like the IFS in medium-range forecasting, while others like MetNet (Sønderby et al., 2020) and NowcastNet (Zhang et al., 2023) have pushed the state of the art in nowcasting. These advances span a diverse set of architectures, including convolutional models (Rasp & Thuerey, 2021), graph neural networks (Keisler, 2022), and Transformers (Pathak et al., 2022a; Nguyen et al., 2023a; Chen et al., 2023c;a). Probabilistic forecasting has also gained traction through methods based on ensembles (Kochkov et al., 2024; Lang et al., 2024) and generative models (Price et al., 2024; Oskarsson et al., 2024; Couairon et al., 2024), which improve the modeling of uncertainty and extreme weather events. These advances have been fueled by open-source datasets and benchmarks. WeatherBench (Rasp et al., 2020; 2023) introduced a benchmark for global medium-range forecasting, with well-defined metrics and a public leaderboard. Subsequent efforts like ChaosBench (Nathaniel et al., 2024) and SubseasonalClimateUSA (Mouatadid et al., 2024) extended this work to subseasonal-to-seasonal prediction. Beyond benchmarks, software libraries such as ClimateLearn (Nguyen et al., 2023b) and Scikit-downscale (Hamman & Kent, 2020) have further streamlined the development of ML models by offering tools for data access, preprocessing, training, and evaluation. Despite progress, most of these efforts have centered on global forecasting.

**Regional weather forecasting efforts**  Regional forecasting has recently gained growing interest within the machine learning community. Hi-LAM (Oskarsson et al., 2023) was among the first to

adapt global models like Graphcast (Lam et al., 2023) to the limited-area setting by incorporating boundary forcing and introducing a hierarchical multi-scale graph structure designed for regional prediction. Diffusion-LAM (Oskarsson et al., 2023) extends this framework by employing denoising diffusion models to capture probabilistic uncertainty in regional forecasts. More recent works such as YingLong-Weather (Xu et al.) and MetMamba (Qin et al., 2024) leverage transformer-based and Mamba (Gu & Dao) architectures, respectively, and apply boundary forcing in a similar fashion to Hi-LAM and Diffusion-LAM. Another complementary line of work incorporates global context directly, conditioning the regional model on coarse-resolution global reanalyses or operational forecasts to improve boundary coherence (Nipen et al., 2024; Pathak et al., 2024).

Despite these advances, there remains a lack of standardization across datasets, model inputs, and evaluation protocols, which limits fair comparison. Specifically, Hi-LAM, Diffusion-LAM, and Nipen et al. (2024) are trained on MEPS (Müller et al., 2017), a regional dataset covering parts of Scandinavia and the Baltics; YingLong-Weather and Stormcast utilize the HRRR dataset (Dowell et al., 2022; James et al., 2022) over the U.S.; and MetMamba uses ERA5 cropped to a regional domain. The most relevant prior effort to ours is BharatBench (Choudhury et al., 2024), which curated a version of IMDAA for regional forecasting over India. However, it supports only coarse (1.08°) resolution, and does not include strong baselines or standardized evaluations.

## 3 DATASET DETAILS

### 3.1 RAW DATA SOURCES

IndiaWeatherBench is built upon the Indian Monsoon Data Assimilation and Analysis (IMDAA) reanalysis dataset (Ashrit et al., 2020), a high-resolution regional reanalysis developed through collaboration between the Indian Ministry of Earth Sciences (MoES), the UK Met Office, and the India Meteorological Department (IMD). IMDAA was designed specifically to support improved understanding and forecasting of the Indian summer monsoon, one of the most complex and economically consequential weather systems. IMDAA employs a 4D-Var data assimilation system integrated within the Met Office Unified Model (UM), which ingests a wide array of observational data including satellite and conventional sources. The full raw dataset includes over 57 meteorological variables across 63 vertical pressure levels, spans the period from 1979 to 2018 (extended to 2020), and offers hourly data at a spatial resolution of 0.12° (approximately 12km), making it one of the highest-resolution publicly available reanalysis datasets for the Indian subcontinent. The fine spatial and temporal granularity of IMDAA makes it a valuable resource for machine learning-based forecasting methods, which demands dense, high-quality training data.

Despite its scientific value, the raw IMDAA dataset presents several challenges for machine learning researchers. First, the data is huge, spanning several terabytes, and downloading the data from its original site (https://rds.ncmrwf.gov.in/) is non-trivial, requiring manual access procedures and resulting in slow transfer speeds. Second, the raw data is stored in formats and conventions designed for meteorological analysis, making it difficult to integrate directly into modern ML pipelines. Third, the dataset lacks standard preprocessing infrastructure required for ML workflows such as data normalization and predefined train-validation-test splits, complicating reproducibility and model comparison. To make the dataset more accessible, IndiaWeatherBench provides a curated and standardized subset of IMDAA optimized for machine learning applications.

### 3.2 INDIAWEATHERBENCH CURATED DATA

The IndiaWeatherBench benchmark includes a curated and preprocessed version of IMDAA that focuses on a spatial domain ranging from 6°N to 36.72°N latitude and from 66.6°E to 97.25°E longitude, corresponding to a $256 \times 256$ grid at the native 0.12° resolution. This area covers the entirety of the Indian subcontinent and surrounding ocean basins that influence monsoon dynamics. We reduce the size of the original data by temporally subsampling the raw data to 6-hour intervals (00, 06, 12, 18UTC), following the practice in WeatherBench 2 (Rasp et al., 2023). IndiaWeatherBench includes 20 years of data, spanning from 2000 to 2019, which we divide into three non-overlapping splits: training (2000–2017), validation (2018), and test (2019), corresponding to approximately 26,500, 1,500, and 1,500 samples, respectively. IndiaWeatherBench includes a total of 43 distinct channels grouped into three categories: single-level variables, pressure-level variables at seven

Table 1: List of variables included in IndiaWeatherBench, grouped by type. Pressure-level variables are provided at seven vertical levels: 50, 250, 500, 600, 700, 850, and 925 hPa.

| Category | Variables |
|---|---|
| **Single-level variables** | TMP (2m temperature) |
| | UGRD (10m U wind), VGRD (10m V wind) |
| | APCP (Total precipitation) |
| | PRMSL (Mean sea level pressure) |
| | TCDCRO (Total cloud cover) |
| **Pressure-level variables** | TMP_prl (Temperature) |
| | HGT (Geopotential height) |
| | UGRD_prl (U wind), VGRD_prl (V wind) |
| | RH (Relative humidity) |
| **Static fields** | MTERH (Terrain height) |
| | LAND (Land cover) |

vertical levels (50, 250, 500, 600, 700, 850, and 925hPa), and static fields. Table 1 shows the full list of variables available in IndiaWeatherBench. One year of data has a size of 16 GB.

To support a variety of machine learning workflows, IndiaWeatherBench supports two data formats: Zarr and HDF5. The Zarr version preserves the full dataset structure in a cloud-friendly, array-based format compatible with tools like Xarray, enabling convenient filtering, slicing, and visualization across multiple variables and dimensions. This format is well-suited for scientific analysis and prototyping. However, since Zarr stores each variable as a separate chunked array, reading multiple variables at arbitrary time steps can be inefficient. To address this, IndiaWeatherBench also provides a more ML-optimized HDF5 version. In this format, the dataset is pre-split into train, val, and test directories, with each file corresponding to a single time step and containing all available variables. This structure enables fast and selective loading of individual samples, reduces memory overhead, and supports efficient batching and parallel data pipelines. The HDF5 format is compatible with conventional data loaders and offers fine-grained control over variable selection and spatial subsetting, making it the preferred choice for deep learning.

## 4 REGIONAL FORECASTING BASELINES

We formulate regional weather forecasting as the task of learning a function $F_\theta$ that maps historical regional weather states and auxiliary information to future forecasts over the region. Let $X_t \in \mathbb{R}^{V \times H \times W}$ denote the high-resolution regional weather state at time $t$, where $H \times W$ is the spatial resolution of the grid and $V$ is the number of meteorological variables. The forecasting model takes as input a history of past states $X_{t-h:t}$ over a window of length $h$, along with auxiliary inputs $S_{t-h:t}$, and predicts the next future state $X_{t+1}$:

$$F_\theta : (X_{t-h:t}, S_{t-h:t}) \longrightarrow \hat{X}_{t+1}. \tag{1}$$

The auxiliary input $S$ provides additional context about the broader atmospheric state beyond the interior regional domain. This information is necessary since regional models only observe a limited area of the full weather system and may otherwise produce inconsistent or inaccurate forecasts due to missing external influences. In practice, $S$ can include high-resolution data at the boundaries of the domain or coarser-resolution forecasts from a global model, which we will present in more detail in Section 4.1. To generate longer forecasts, we apply the model autoregressively, repeatedly feeding back the predicted state $\hat{X}_{t+1}$ as input in the next step until we reach the target lead time.

### 4.1 BOUNDARY CONDITIONING STRATEGIES

To account for the influence of atmospheric dynamics outside the regional domain, we explore two distinct boundary conditioning strategies for regional forecasting. The first strategy, known as boundary forcing, incorporates high-resolution data at the spatial boundaries of the region. In this approach, the auxiliary information $S_t$ represents the surrounding pixels that lie just outside the region of interest at each time step $t$. We can wrap these boundary values $S_t$ around the current regional

state $X_t$ to provide a single input to the model with better continuity of meteorological fields across domain edges. This method is commonly used in existing data-driven methods (Oskarsson et al., 2023; Larsson et al., 2025; Xu et al.) and aligns well with numerical weather prediction practices. However, it requires the boundary information to be available at the same spatial resolution as the regional model. In operational settings, this is only feasible if a global forecasting model exists at high resolution, an assumption that may not hold for many regions due to computational cost.

The second strategy conditions the model on coarse-resolution global forecasts from existing operational systems (e.g., IFS, GFS, Graphcast) (Nipen et al., 2024; Pathak et al., 2024). In this approach, $S_t$ is a lower-resolution view of the global atmospheric state, which is cropped to the region of interest with possibly surrounding pixels. We then interpolate the coarse-resolution input $S_t$ to match the grid size of $X_t$ and concatenate them to form a single input to the model. This setup enables learning-based fusion of interior and global context, allowing the model to account for synoptic-scale drivers while preserving fine-scale variability. This strategy is highly applicable in real-world deployments, where coarse global forecasts are readily available but high-resolution boundary values are not. However, it requires the forecasting model to effectively integrate information from two distinct sources – interior history and external global context, which can increase model complexity and training difficulty.

We note that in an operational setting, the auxiliary input $S_t$ would typically be provided by a global forecasting model. However, to simplify the benchmark setup and isolate the influence of the global model, we use the ground-truth weather state for $S_t$ during training and evaluation. This means using the true boundary pixel values in the case of boundary forcing, and the true low-resolution global state in the case of coarse-resolution conditioning.

## 4.2 NEURAL NETWORK ARCHITECTURES

We establish a strong set of baselines in IndiaWeatherBench, spanning convolutional, transformer, and graph neural network architectures. Note that for Stormer and Graph-based models, we only use their architectures and not their pretrained models.

**UNet** The UNet architecture was originally developed for biomedical image segmentation (Ronneberger et al., 2015). The model has a symmetric encoder-decoder structure with skip connections to retain spatial information across different scales. UNet has proven effective in dense prediction tasks in computer vision, making it a simple yet strong baseline for high-resolution regional forecasting.

**Transformer-based models** We consider three state-of-the-art transformer architectures: FourCast-Net (Pathak et al., 2022b), Pangu-Weather (Bi et al., 2022), and Stormer (Nguyen et al., 2023c). The three architectures differ mainly in the embedding layer and the transformer backbone. FourCastNet and Pangu-Weather both use a simple linear patch embedding layer, while Stormer employs a cross-attention embedding module that captures non-linear interactions between different input variables. For the transformer backbone, FourCastNet uses Adaptive Fourier Neural Operator (AFNO) (Guibas et al., 2021) that interleaves channel and spatial mixing, whereas Pangu-Weather uses a 3D version of Swin transformer (Liu et al., 2022), and Stormer uses a standard transformer backbone.

**Graph-based models** We include GraphCast, a graph neural network model originally developed for global weather forecasting (Lam et al., 2023). Graphcast encodes atmospheric states onto the nodes of a multi-scale mesh graph, where each node represents a spatial location and each edge captures spatial interactions. The graph is constructed by merging multiple levels of icosahedral meshes, allowing the model to propagate information over both short and long distances. This multi-scale structure enables GraphCast to capture meteorological phenomena across a wide range of spatial scales. The Hierarchical Graph Neural Network (Hi) (Oskarsson et al., 2023) extends Graphcast by replacing the merged mash with a level-wise hierarchy. By connecting different mesh resolutions through vertical edges, Hi allows more structured and directional information flow from fine to coarse and vice versa. This hierarchical design reduces artifacts observed in Graphcast and enhances the model's ability to integrate local details with broader spatial context, making it especially suitable for regional forecasting tasks (Oskarsson et al., 2023).

## 4.3 TRAINING OBJECTIVES

In this benchmark, we adopt a dynamics learning formulation, where the model learns to predict the *increment* between future and current states $\Delta X_{t+1} = X_{t+1} - X_t$ rather than directly outputting

the next state $X_{t+1}$. During evaluation, we can obtain the actual next-state forecast by adding the predicted increment to the initial condition: $\hat{X}_{t+1} = X_t + \hat{\Delta} X_{t+1}$. This formulation follows the practice in state-of-the-art models like GraphCast and Stormer, and has proven more effective than next-state prediction. IndiaWeatherBench supports both deterministic and probabilistic forecasting.

**Deterministic prediction** For deterministic forecasting, we minimize the latitude-weighted mean squared error between the predicted and ground-truth state increments. Let $\theta$ denote the model parameters and $\Delta X_{t+1}$ the true increment. The loss is defined as:

$$\mathcal{L}_{\text{deter}}(\theta) = \frac{1}{VHW} \sum_{v=1}^{V} \sum_{i=1}^{H} \sum_{j=1}^{W} L(i) \left\| \hat{\Delta} X_{t+1}^{vij} - \Delta X_{t+1}^{vij} \right\|_2^2, \tag{2}$$

where $L(i) = \frac{\cos(\text{lat}(i))}{\frac{1}{H} \sum_{i'=1}^{H} \cos(\text{lat}(i'))}$ is a weighting function based on the latitude of row $i$ to account for the non-uniformity of gridding the spherical globe.

**Probabilistic modeling.** To model the uncertainty in regional dynamics, we adopt denoising diffusion models from the EDM (Elucidated Diffusion Model) framework (Karras et al., 2022). These models learn the conditional distribution $p_\theta(\Delta X_{t+1} \mid X_{t-h:t}, S_{t-h:t})$ by reversing a predefined noising process. During training, we corrupt the true increment $\Delta X_{t+1}$ with Gaussian noise and train the model to predict the clean signal from its noisy version using a score-based objective:

$$\mathcal{L}_{\text{prob}}(\theta) = \mathbb{E}_{t,\epsilon} \left[ \left\| \epsilon - \hat{\epsilon}_\theta(\Delta X_{t+1}^{(s)}, X_{t-h:t}, S_{t-h:t}) \right\|_2^2 \right], \tag{3}$$

where $\Delta X_{t+1}^{(s)}$ is the noisy increment at noise level $s$, and $\epsilon$ is the injected noise. The model learns to denoise $\Delta X_{t+1}^{(t)}$ by estimating the noise $\hat{\epsilon}_\theta$ from the conditioning inputs. During inference, forecasts are generated by sampling from the learned distribution using a reverse-time stochastic differential equation (SDE). The EDM framework enables automatic tuning of sampling hyperparameters and offers strong mode coverage for complex weather dynamics.

Together, these two training paradigms provide complementary capabilities: deterministic models are fast and interpretable, while diffusion-based models provide calibrated probabilistic forecasts that are essential for downstream risk-sensitive applications.

### 4.4 EVALUATION METRICS

To comprehensively assess model performance, we evaluate both the point prediction accuracy and the probabilistic calibration of forecasts. Our benchmark supports four primary evaluation metrics: Root Mean Square Error (RMSE), Anomaly Correlation Coefficient (ACC), Continuous Ranked Probability Score (CRPS), and Spread/Skill Ratio (SSR). We detail these metrics in Appendix C.2.

## 5 EXPERIMENTS

We conduct extensive experiments to demonstrate the capabilities and flexibility of IndiaWeather-Bench as a benchmark for regional weather forecasting. We train and evaluate four representative architectures – UNet, Stormer, GraphCast, and Hi, under different boundary conditioning strategies and training objectives. Our evaluation covers both overall forecasting accuracy and performance under extreme weather conditions. Due to space constraints, we focus on the deterministic forecasting results in the main text and defer the discussion of probabilistic forecasting results to Appendix D.4. We additionally compare deep learning baselines with climatology in Appendix D.2.

**Boundary conditioning details.** For the boundary forcing strategy, we use a 10-pixel-wide boundary around the regional domain at each time step. These boundary values are extracted from the ground truth and wrapped around the interior regional state $X_t$ to form a single input tensor. For the coarse-resolution conditioning strategy, we use ERA5 (Hersbach et al., 2020) data as the external low-resolution input during training. Specifically, for each time step, we crop ERA5 to cover the Indian region, resulting in a $124 \times 124$ grid, and then bilinearly interpolate it to match the $256 \times 256$ resolution of IndiaWeatherBench. We use the same set of variables for both the regional and ERA5 inputs and concatenate them along the channel dimension. At test time, we also consider replacing ERA5 with global forecasts from IFS to mimic an operational setting.

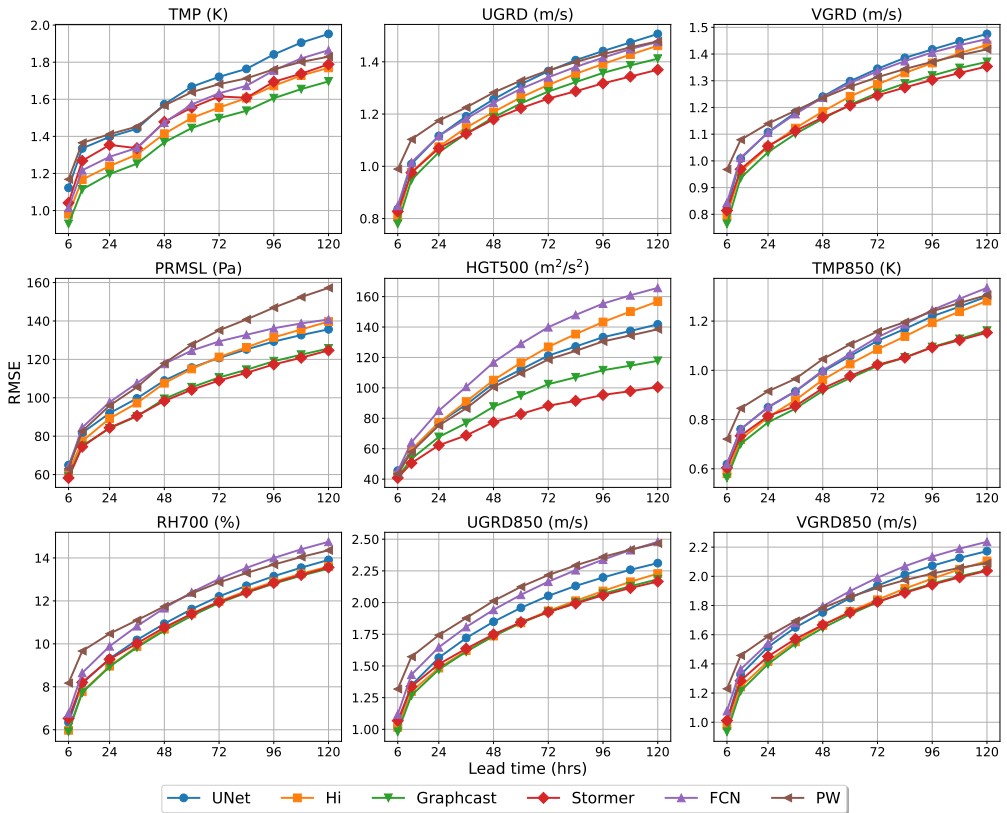

Figure 1: Performance of baselines with boundary forcing across 9 key variables.

**Training and evaluation details.** We constrain the total parameter count of each baseline model to between 25M and 35M to ensure a fair comparison across architectures. Please refer to Appendix C.1 for the complete hyperparameters of the baselines. We train all models using a consistent set of 39 input channels, which includes temperature at 2 meters, the u and v components of wind at 10 meters, mean sea level pressure, and five pressure-level variables – geopotential height, temperature, u-wind, v-wind, and relative humidity, each provided at seven vertical levels. We follow the standard data splits defined in Section 3.2, and train each model for 100 epochs with a batch size of 32. We use AdamW (Kingma & Ba, 2014) optimizer with a base learning rate of $2e - 4$, using a 10-epoch linear warmup, followed by a cosine decay schedule for the remaining 90 epochs. For model selection, we evaluate the validation loss after each training epoch and use the model with the lowest validation loss for testing. We use RMSE as the evaluation metric, and refer readers to Appendix D.3 for additional metrics. We keep the same training and evaluation setting across all experiments.

## 5.1 BENCHMARK RESULTS

Figure 1 shows that under the boundary forcing setting, Stormer and Graphcast achieve the best overall performance across most variables and lead times, consistent with their strong performance in global weather forecasting. On the other hand, FCN and Pangu-Weather lag behind, indicating that prior results in global forecasting may not directly translate to the regional setting. Hi, despite being proposed as an improved hierarchical extension of Graphcast, underperforms its predecessor across all variables. UNet performs competitively and is often within a small margin of the top performers. While not designed specifically for weather forecasting, its simplicity and robustness make it a strong baseline for high-resolution regional prediction.

In contrast, Figure 2 shows that under coarse-resolution conditioning, the ranking of methods shifts significantly[1]. Most notably, Stormer becomes the worst-performing model, with forecasting error

---

[1]We ran into numerical instabilities with FCN, and thus did not include it in this setting.

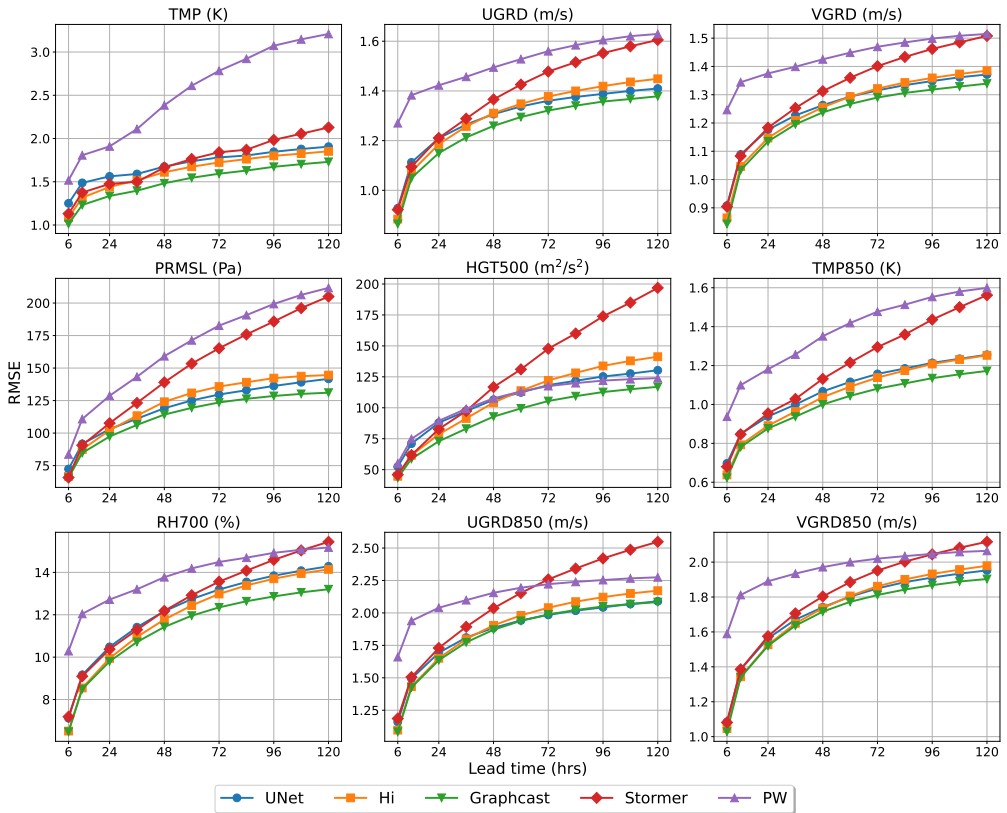

Figure 2: Performance of baselines with coarse-resolution conditioning across 9 key variables.

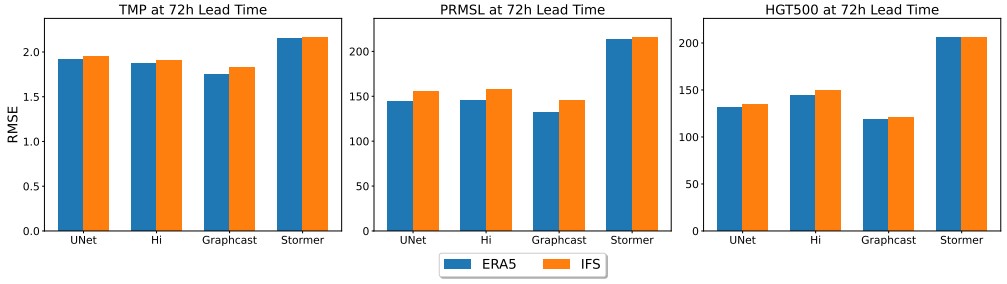

Figure 3: Comparison of the two types of global data used for coarse-resolution conditioning.

growing rapidly over time across all variables. We hypothesize that this degradation stems from an incompatibility between Stormer's input tokenization scheme and the coarse-resolution conditioning strategy. Specifically, we interpolate the global ERA5 input to the same spatial resolution as the regional data and concatenate it along the channel dimension. Stormer then tokenizes this combined input into patches, such that each token blends high-resolution regional context with upsampled coarse global input. This mixing of incompatible spatial scales within each token likely disrupts the attention mechanism, leading to poor generalization.

In the above experiment, we used the future ground-truth ERA5 data as the coarse-resolution conditioning for the model during rollout, which is not realistic in an operational setting. To simulate a real-world scenario, we replaced the ERA5 data with forecasts from the Integrated Forecasting System (IFS) (Wedi et al., 2015). As shown in Figure 3, this change leads to a slight degradation in performance for all models, which is expected due to the distribution shift between ERA5 reanalysis and IFS forecasts. Crucially, the performance gap is small, demonstrating that our coarse-resolution conditioning strategy is robust for operational deployment.

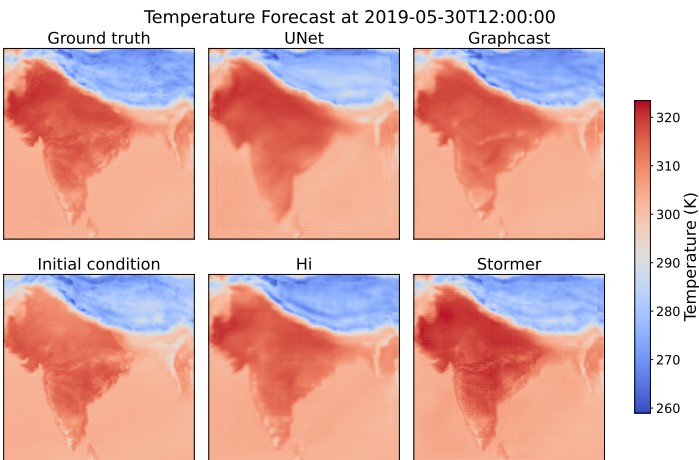

Figure 4: 5-day temperature forecasts of different models initialized at 12UTC, 2019-05-25.

## 5.2 EXTREME WEATHER EVENTS

We evaluate the performance of different models during a record-breaking heatwave event in India that occurred from May 25 to June 1, 2019. Figure 4 visualizes the 5-day temperature forecasts from different models initialized at 12:00 UTC on May 25 and evaluated at 12:00 UTC on May 30. While all models roughly capture the spatial pattern of surface temperature, there are notable differences in accuracy and bias. Hi appears to produce the most realistic forecast, closely matching the ground truth over Central and Northern India. Graphcast underestimates the temperature, particularly in Central India. In contrast, Stormer overestimates the temperature in large parts of the domain, producing overly hot forecasts that deviate from observed values.

These trends are consistent in Figure 5, which shows the average predicted temperature over Central India compared to the reference data at 12UTC for each day between May 25 and June 1. Stormer and UNet exhibit a strong warm bias throughout the period, consistently overshooting the observed temperature, while Graphcast shows a persistent cold bias. Notably, Hi tracks the temporal trend of the observed temperature well and maintains a small error across the forecast horizon, highlighting its potential advantage in predicting extreme events. These results demonstrate that extreme events pose unique challenges and that model behavior can vary substantially under rare conditions.

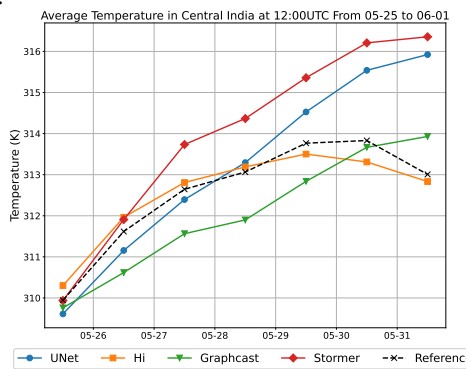

Figure 5: Avg. predicted and reference temperature in Central India from 05-25 to 06-01.

## 6 CONCLUSION

We introduced IndiaWeatherBench, a standardized dataset and benchmark for regional weather forecasting over India. Built on the high-resolution IMDAA reanalysis, IndiaWeatherBench provides a curated, ML-ready dataset along with diverse baselines spanning convolutional, transformer, and graph-based architectures. Our benchmark supports multiple boundary conditioning strategies and training objectives, enabling systematic comparisons under standard and extreme weather conditions.

**Limitations and Future Work** The current benchmark results do not include evaluation on precipitation, an important variable for weather forecasting. Future work can extend IndiaWeatherBench along three axes: (1) data – by incorporating more regional domains in addition to India, (2) models – by including more advanced approaches specialized to regional forecasting, and (3) evaluations – by supporting targeted metrics and validation protocols for precipitation, an important aspect of weather forecasting for India.

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

## A    LICENSES AND TERMS OF USE

We developed IndiaWeatherBench using the data from IMDAA, which belongs to the NCMRWF, Ministry of Earth Science, Government of India. IMDAA is available under the CC BY-NC-SA 4.0 license (`https://rds.ncmrwf.gov.in/privacy`).

## B    BROADER IMPACTS

IndiaWeatherBench aims to advance the scientific and practical capabilities of regional weather forecasting, with a specific focus on high-impact and climate-sensitive regions such as India. Accurate regional forecasts are crucial for agriculture, disaster preparedness, water resource management, and public health, especially in countries with large populations and vulnerable infrastructure. By standardizing datasets, baselines, and evaluation protocols, IndiaWeatherBench enables reproducible research, lowering the barrier for broader participation in atmospheric science from the machine learning community. We encourage responsible and open use of this benchmark, and we release all code and data under permissive licenses to foster accessibility and transparency.

## C    BENCHMARK DETAILS

### C.1    BASELINE ARCHITECTURE DETAILS

For reproducibility and fair comparisons across architectures, we kept the parameter count of each architecture from 30 to 35 million. Table 2, 3, 4, 5 show the exact hyperparameters we used for each architecture.

Table 2: Default hyperparameters of UNet

| Hyperparameter | Meaning | Value |
| --- | --- | --- |
| Hidden channels | Base number of hidden channels | 64 |
| Channel multipliers | Channel multipliers per resolution stage | [1, 2, 4] |
| Blocks per level | Number of convolutional blocks per level | 2 |
| Use mid attention | Use attention in the bottleneck | False |

Table 3: Default hyperparameters of GraphCast

| Hyperparameter | Meaning | Value |
| --- | --- | --- |
| Hidden size | Hidden dimension for node features | 512 |
| MLP layers | Number of layers in node MLP | 1 |
| Processor layers | Number of graph message-passing layers | 16 |
| Aggregation type | Aggregation method for messages | Sum |

Table 4: Default hyperparameters of Hierarchical GraphCast

| Hyperparameter | Meaning | Value |
| --- | --- | --- |
| Hidden size | Hidden dimension for node features | 128 |
| MLP layers | Number of layers in node MLP | 1 |
| Processor layers | Number of graph message-passing layers | 16 |

Table 5: Default hyperparameters of Stormer

| Hyperparameter | Meaning | Value |
| --- | --- | --- |
| Patch size | Size of image patches | 2 |
| Hidden size | Embedding dimension | 512 |
| Depth | Number of transformer layers | 8 |
| Attention heads | Number of self-attention heads | 8 |

## C.2 EVALUATION METRICS

IndiaWeatherBench supports 4 standard metrics: Root Mean Square Error (RMSE) and Anomaly Correlation Coefficient (ACC) for forecast accuracy, and Continuous Ranked Probability Score (CRPS) and Spread/Skill Ratio (SSR) for probabilistic forecast calibration. In all metrics below, we denote $X$ and $\tilde{X}$ as the ground truth and forecast, respectively. We use $H$ and $W$ to denote the latitude and longitude dimensions, respectively. We present the metrics for a single data point and a single variable.

**Root Mean Square Error (RMSE).** RMSE is a standard metric for point forecasting that measures the average squared difference between the predicted and true values. To account for the uneven surface area of latitude-longitude grids, we apply latitude weighting:

$$\text{RMSE} = \sqrt{\frac{1}{H \times W} \sum_{i=1}^{H} \sum_{j=1}^{W} L(i) \left( \tilde{X}_{i,j} - X_{i,j} \right)^2}, \tag{4}$$

where $L(i)$ is a latitude-based weighting function proportional to $\cos(\phi_i)$, and $\phi_i$ is the latitude of grid row $i$. RMSE captures the overall forecast accuracy at each grid point.

**Anomaly Correlation Coefficient (ACC).** ACC evaluates the spatial correlation between forecast anomalies and ground-truth anomalies with respect to a climatological mean:

$$\text{ACC} = \frac{\sum_{i,j} L(i) \tilde{X}_{i,j}^{'} X_{i,j}^{'}}{\sqrt{\sum_{i,j} L(i) \tilde{X}_{i,j}^{'2} \sum_{i,j} L(i) X_{i,j}^{'2}}}, \tag{5}$$

where $\tilde{X}^{'} = \tilde{X} - C$ and $X^{'} = X - C$, with $C$ denoting the climatology computed as the temporal mean of the ground truth over a fixed historical window. We refer to Appendix D.2 for details on climatology calculation.

**Continuous Ranked Probability Score (CRPS).** CRPS measures the quality of probabilistic forecasts by quantifying the distance between the predicted cumulative distribution function (CDF) and the ground-truth observation. Following prior work, we use the following formulation:

$$\text{CRPS} = \mathbb{E}_{x \sim p_\theta} \left[ |x - X| \right] - \frac{1}{2} \mathbb{E}_{x,x' \sim p_\theta} \left[ |x - x'| \right], \tag{6}$$

where $p_\theta$ is the model's predictive distribution. The first term captures forecast error, while the second term penalizes overdispersion. We note that both terms are latitude-weighted by $L(i)$, which we omit in the formulation for simplicity. Lower CRPS values indicate better-calibrated forecasts.

**Spread/Skill Ratio (SSR).** SSR compares ensemble spread to forecast skill. A well-calibrated ensemble should have a spread that matches its error. We first compute the average ensemble spread:

$$\text{Spread} = \sqrt{\frac{1}{H \times W} \sum_{i=1}^{H} \sum_{j=1}^{W} L(i) \text{Var}_m[X_{i,j}]} \tag{7}$$

where $\text{Var}_m$ denotes the variance in the ensemble dimension. We then define SSR as:

$$\text{SSR} = \frac{\text{Spread}}{\text{RMSE}_{\text{ens}}}, \tag{8}$$

where $\text{RMSE}_{\text{ens}}$ is the RMSE of the ensemble mean. An SSR close to 1 indicates a well-calibrated ensemble, while values significantly above or below 1 indicate over- or underdispersion.

# D ADDITIONAL RESULTS

## D.1 COMPARING DIFFERENT BOUNDARY CONDITIONING STRATEGIES

Figure 6 compares the performance of different baselines when using the two boundary conditioning strategies. UNet, Graphcast, and Hi perform comparably or slightly better with coarse-resolution conditioning relative to boundary forcing, but Stormer degrades noticeably. These results align with our main results, and emphasize the importance of aligning architectural design with boundary conditioning strategy, since what works well under one setup may fail under another.

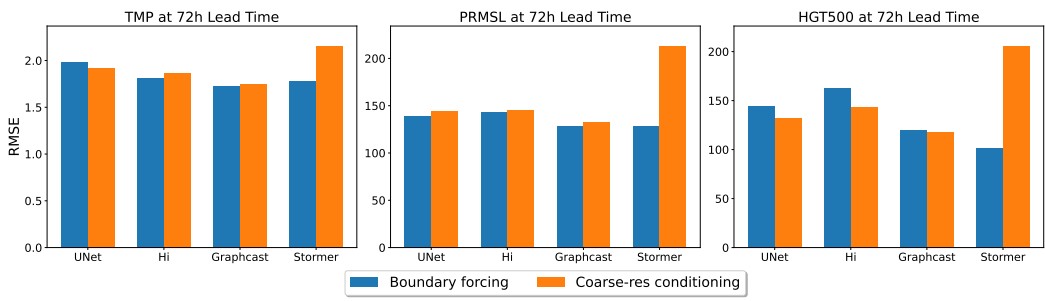

Figure 6: Comparison of the two boundary conditioning strategies with different architectures across 3 key variables at 72-hour lead time.

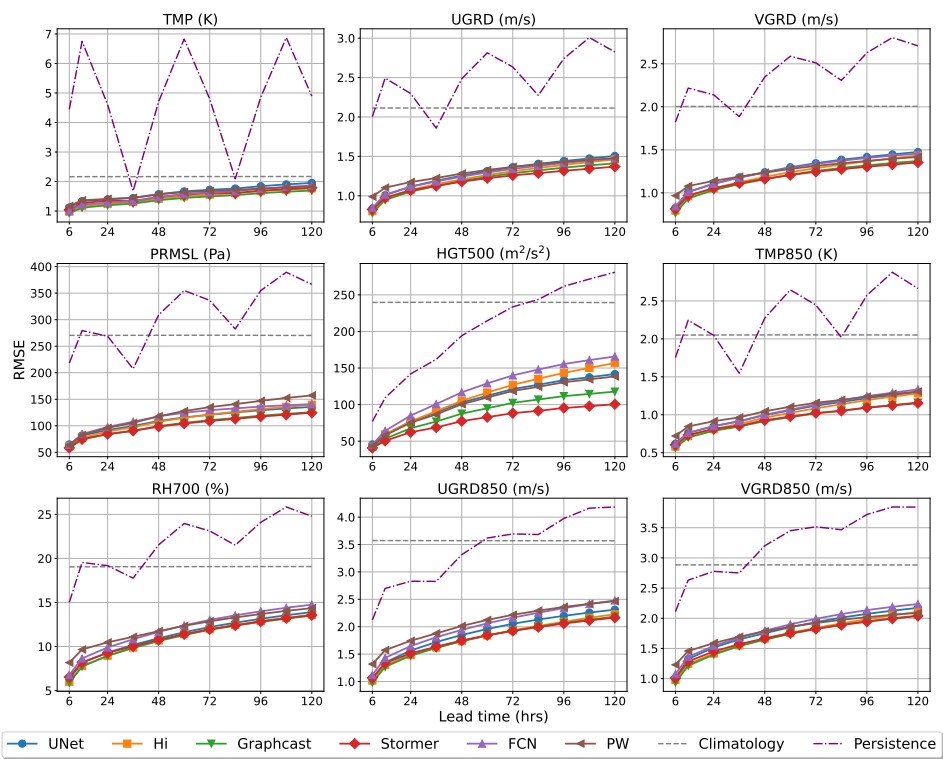

Figure 7: RMSE of deep learning baselines with boundary forcing vs persistence and climatology.

## D.2 MAIN RESULTS WITH CLIMATOLOGY AND PERSISTENCE

We compare the deep learning methods with climatology and persistence, two simple baselines commonly used in weather forecasting, to better evaluate their forecast skills. We calculate climatology by taking the mean value of each time across the training set and predicting that to be the forecast for the test year 2019. This means that for a particular day and time (e.g., December 4, 6:00 UTC), the forecast is the mean of 18 values for the years 2000-2017 for that date and time.

## D.3 ADDITIONAL METRICS

Figures 9 and 10 show the ACC score of the 4 deep learning baselines with two different boundary conditioning strategies.

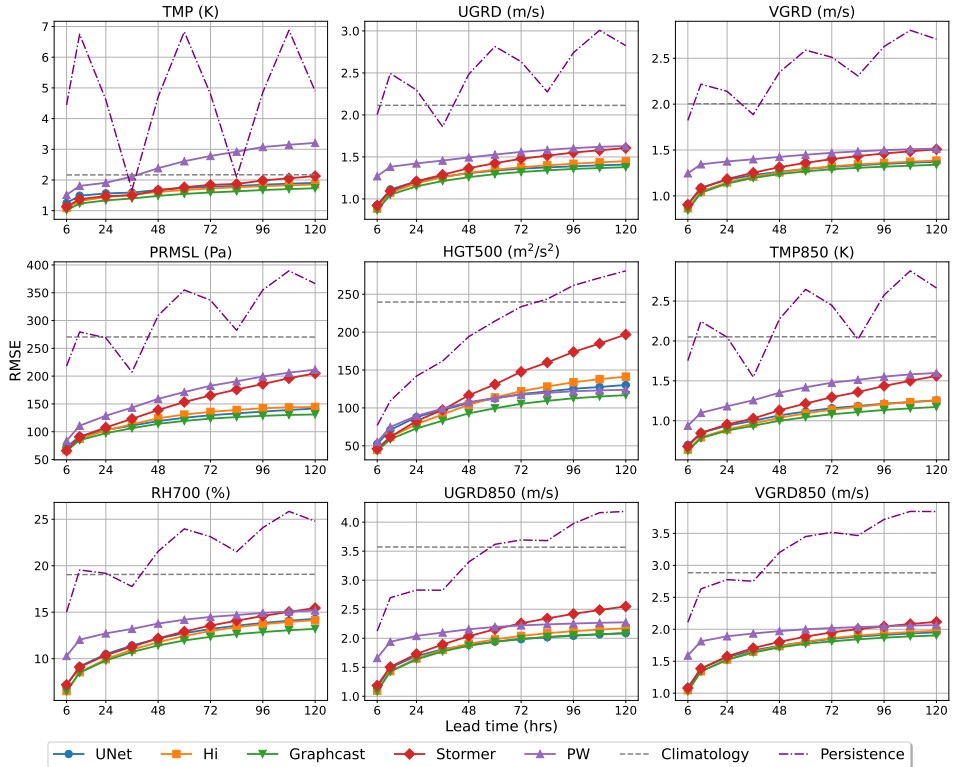

Figure 8: RMSE of deep learning baselines with coarse conditioning vs persistence and climatology.

## D.4 PROBABILISTIC FORECASTING

In addition to deterministic forecasting, IndiaWeatherBench also supports probabilistic forecasting with diffusion models. We followed the diffusion formulation in Graphcast, which we refer to the original paper (Lam et al., 2023) and Karras et al. (2022) for more details. We trained the diffusion model using the same training and optimization details as the deterministic models. After training, we sampled from the model using DPMSolver++2S (Lu et al., 2022) with sampling hyperparameters specified in Table 6.

Table 6: Noise schedule hyperparameters

| Name | Notation | Value, sampling | Value, training |
|------|----------|-----------------|-----------------|
| Number of ensemble members | $N$ | 50 | – |
| Maximum noise level | $\sigma_{max}$ | 80 | 88 |
| Minimum noise level | $\sigma_{min}$ | 0.03 | 0.02 |
| Shape of noise distribution | $\rho$ | 7 | 7 |
| Number of noise levels | $N$ | 20 | 20 |
| Stochastic churn rate | $S_{churn}$ | 2.5 | 2.5 |
| Churn maximum noise level | $S_{max}$ | 80 | 80 |
| Churn minimum noise level | $S_{min}$ | 0.75 | 0.75 |
| Noise level inflation factor | $S_{noise}$ | 1.05 | 1.05 |

Given limited time and resources, we only benchmark UNet and Stormer with boundary forcing for probabilistic forecasting. Figures 11 and 12 show the performance of the two models using CRPS and SSR as the metric, respectively. The SSR score shows that the model is under-dispersive in almost all variables except for TMP, with Stormer being more severe. Future work can explore various ways to improve the probabilistic framework, including but not limited to better diffusion training, adding

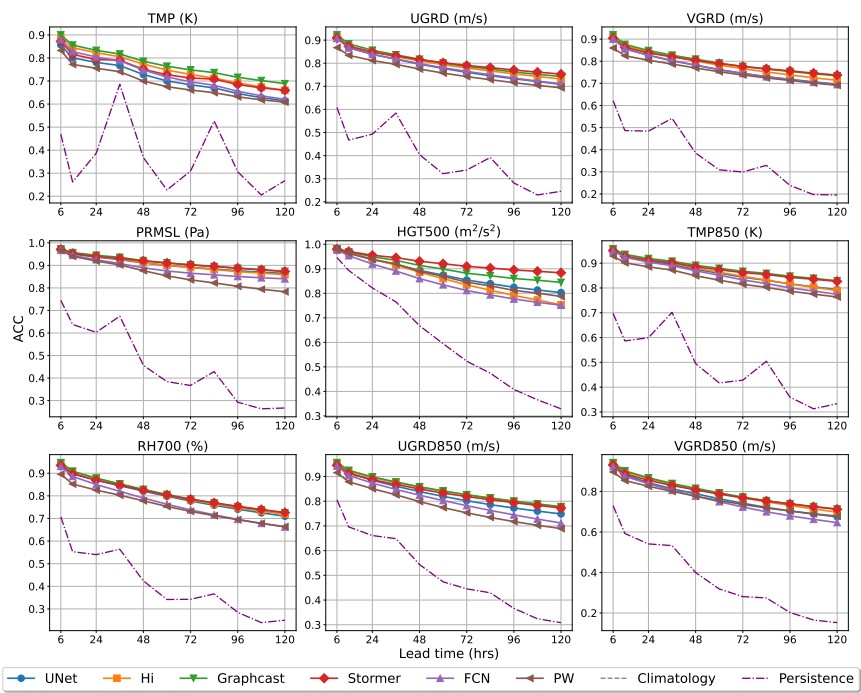

Figure 9: ACC of deep learning baselines with boundary forcing vs persistence and climatology.

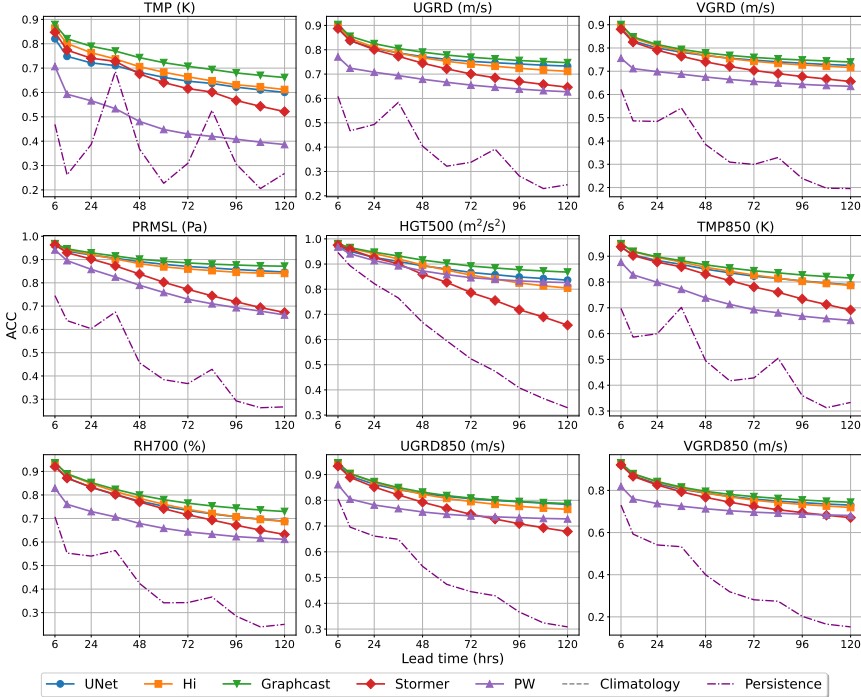

Figure 10: ACC of deep learning baselines with coarse conditioning vs persistence and climatology.

random noise to the initial conditions to improve dispersion, or using the ERA5 Ensemble of Data Assimilations (EDA) (Isaksen et al., 2010) for initial conditions.

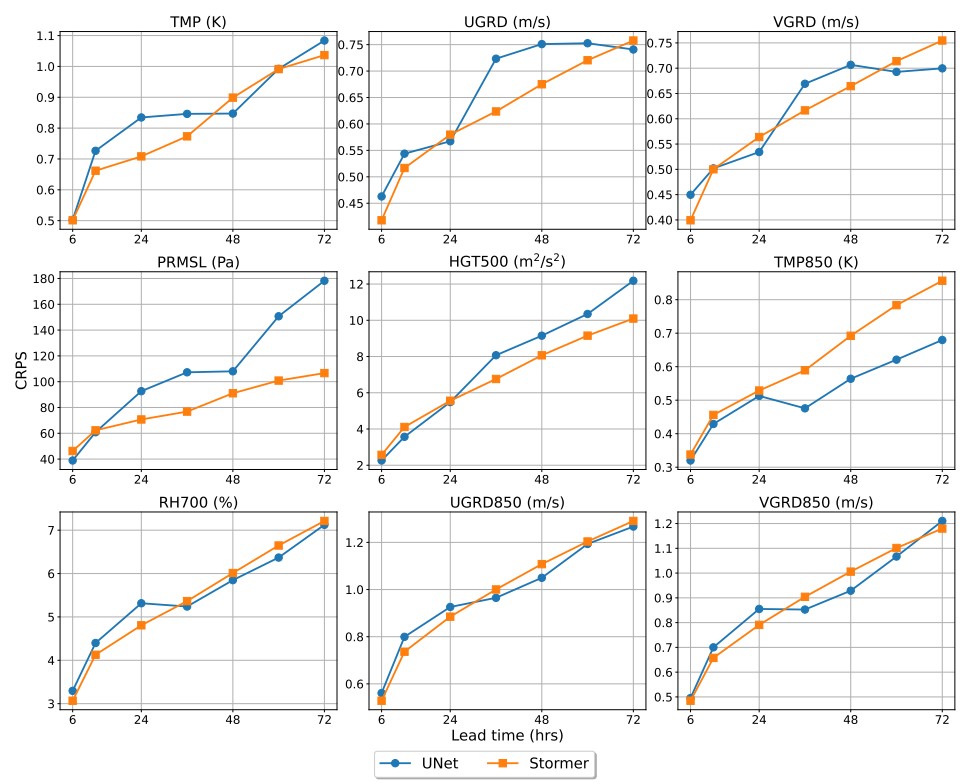

Figure 11: CRPS performance of UNet+diffusion with boundary forcing for probabilistic forecasting.

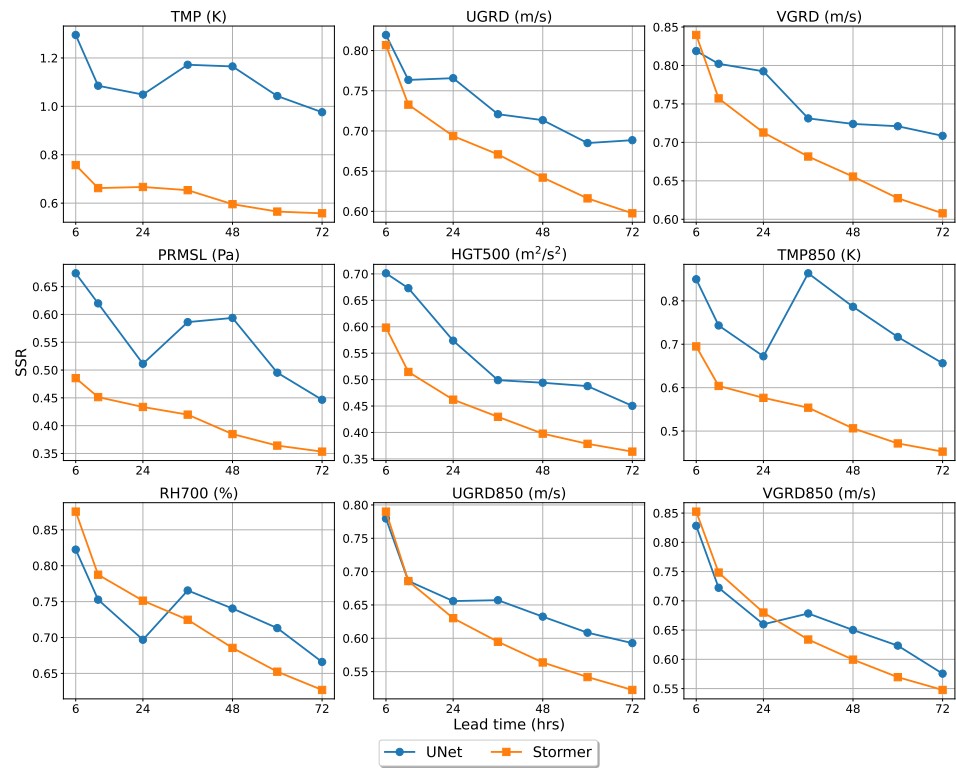

Figure 12: SSR performance of UNet+diffusion with boundary forcing for probabilistic forecasting.

