# OpenReview forum: "IndiaWeatherBench: A Dataset and Benchmark for Data-Driven Regional Weather Forecasting over India"
_ICLR.cc/2026/Conference — Submitted to ICLR 2026_

### Official Review · Reviewer_m8rS · 2025-10-22

**Soundness:** 2
**Presentation:** 2
**Contribution:** 1
**Rating:** 2
**Confidence:** 4

**Summary:**

The authors propose IndiaWeatherBench as a dataset for regional weather forecasting over India. The goal is to provide a benchmark dataset that enables fair comparisons of different model architectures used in the literature. To this end, the authors employ four architectures and introduce two boundary conditioning strategies for empirical evaluation.

**Strengths:**

- The motivation of the work is clear, including the choice of focusing on India and addressing the lack of standardization across datasets, models, and evaluation methods.
- The literature review on regional and global weather forecasting datasets and models is extensive and up to date, providing a clear understanding of the current state of the field.
- To make the dataset easier to analyze and use, the authors provide the IndiaWeatherBench dataset in two different formats tailored to specific use cases.
- The downstream forecasting task, evaluated using various models from the literature, provides a solid understanding of the dataset’s overall structure and components. The focused analysis on extreme weather events (Section 5.2) is particularly promising and encourages further study and evaluation with additional models.

**Weaknesses:**

- The paper’s novelty appears limited. Many regional and global weather datasets have been proposed recently, so it is unclear what makes this dataset particularly valuable for further research and evaluation. The only clear advantage seems to be the extreme weather forecasting results presented at the end of the experiments, which could hold specific interest.
- The authors mention a similar dataset that covers India for regional forecasting purposes. However, the evaluation of IndiaWeatherBench does not sufficiently justify why the proposed dataset is a better choice. The authors have not effectively highlighted the advantages of their dataset clearly. Section 3 should serve as the core of the manuscript, clearly explaining why other researchers should use this dataset in their experiments, but it currently lacks compelling motivation or interest. The lack of sufficient evaluation of state-of-the-art algorithms on the BharatBench dataset alone does not provide strong justification. Maybe a direct comparison between the two datasets would be more useful and informative.
- The authors claim that one of their motivations is the lack of standardization across datasets, model inputs, and evaluation protocols. While they provide numerical forecasting results for a few models from the literature, this effort does not adequately address the standardization gap they aim to solve. As shown in Section 5 and Section C.1, evaluating only four models with default hyperparameters and standard optimization settings is not convincing as a benchmark study. Moreover, the extent of the hyperparameter search space for each model is not specified.
- I would not consider this a direct weakness, but the strengths highlighted by the authors seem rather standard and common across similar studies and datasets. For example, the authors mention that IndiaWeatherBench is built upon the IMDAA regional reanalysis dataset. In Section 3.1, they note that one of the main issues with the raw IMDAA data is its large size and the need for extensive preprocessing before it can be used for machine learning tasks. However, such preprocessing is a common practice, as most raw datasets require significant data engineering. Beyond providing the curated dataset derived from IMDAA, the rest of the manuscript feels routine and includes elements that are generally expected in similar works.

**Questions:**

- See the weaknesses section for my detailed comments.

---

### Official Review · Reviewer_5zhr · 2025-10-31

**Soundness:** 2
**Presentation:** 2
**Contribution:** 1
**Rating:** 2
**Confidence:** 5

**Summary:**

This paper proposes a regional benchmark for the Indian area. It preprocesses the raw data into a small subset. However, this data is not proposed by the author, they just carried out some engineering work. And the baselines are too old.

**Strengths:**

Perhaps it preprocesses a indian’s meteorological into two formats (Zarr and HDF5).

**Weaknesses:**

1. Different from Weatherbench 2, which provides a comprehensive benchmark for some famous ML-based weather forecasting models published in top journals such as Nature/Science. The baselines in IndiaWeatherBench are too old. The newest baseline included in IndiaWeatherBench is GraphCast, which was published in 2023. Further, the results of Graphcast are retrained by the author. And some baseline like UNet is not designed for weather forecasting. Some ML-based models target for regional weather forecasting are not be examined in this benchmark.
2. Preprocessing the raw data into a subset is not a scientific work, it just some dirty work. As for the data normalization, it just a simple step for this research area. And different from the raw ERA5 data is hundreds of TB, the raw size of IMDAA is only too small. So, the user can download it using more convenient ways, rather than using your proposed IndiaWeatherBench.
3. Overall, this work did not meet the standards for acceptance by ICLR. Overall, it just did some dirty work, such as data preprocessing, and the author exaggerated their contribution.

**Questions:**

See Weaknesses

---

### Official Review · Reviewer_ARLC · 2025-11-01

**Soundness:** 4
**Presentation:** 4
**Contribution:** 2
**Rating:** 4
**Confidence:** 5

**Summary:**

This paper introduces IndiaWeatherBench (IWB), the first standardized benchmark for data-driven regional weather forecasting over India. Built on the IMDAA regional reanalysis (12 km, hourly, 1979–2020), the dataset provides 20 years of preprocessed multi-channel atmospheric fields with train/validation/test splits and supports both deterministic and probabilistic forecasts.
The benchmark compares four main architectures—UNet, GraphCast, Stormer, and Hi-GraphCast—under two boundary conditioning strategies:
(1) Boundary forcing (using fine-resolution edges), and
(2) Coarse-resolution conditioning (using downsampled ERA5 or IFS).
Results show that different architectures respond differently to boundary conditioning (e.g., Stormer degrades under coarse-resolution inputs). The benchmark also includes a case study on the 2019 Indian heatwave.

**Strengths:**

1. First India-specific regional ML benchmark:
Establishes a reproducible, high-resolution (12 km) dataset derived from IMDAA with standardized preprocessing and splits. This fills a critical regional gap between global efforts (WeatherBench 2, ChaosBench) and smaller, bespoke LAM setups.

2. Comprehensive experimental setup:
The authors evaluate multiple strong architectures (GraphCast, Stormer, Pangu-Weather, UNet, Hi-GraphCast) under consistent compute budgets (25–35 M params), and across deterministic and probabilistic formulations—rare in regional forecasting.

3. Practical boundary conditioning investigation:
Two realistic strategies are systematically compared—this is an operationally relevant issue for regional ML models and provides useful insights.

4. Case study on extreme events:
The heatwave experiment highlights model biases and strengths under rare conditions, suggesting that IWB could support impact-based forecasting research.

**Weaknesses:**

1. Ambiguity in IMDAA usage:
While IWB is based on IMDAA, the text sometimes implies IMDAA is used as both the dataset source and the evaluation ground truth. Clarification is needed:
– Is IMDAA directly the “truth” for RMSE/ACC computation, or is it merely the preprocessed source for IWB?
– When ERA5 or IFS are used as conditioning inputs, are comparisons made against IMDAA or within the resampled domain?

2. Weak interpretability of results:
Figures 1–2 show variable-wise RMSE curves, but no unified insight is drawn. Why do some models perform better under one conditioning but not another? What physical patterns explain Stormer’s degradation? A statistical or qualitative analysis would strengthen the conclusions.

3. Incomplete variable coverage:
Precipitation is omitted from evaluation despite being among the most societally critical targets in India. This omission limits the benchmark’s immediate practical utility for hydrological or monsoon-related applications.

4. Lack of generalization discussion:
The benchmark is India-specific, but the paper claims extensibility to other regions without evidence. A small pilot (e.g., SE Asia subset) would make this claim more convincing.

**Questions:**

1. The paper’s core contribution seems primarily infrastructural (dataset and benchmark release), and it is unclear what new scientific discovery or methodological innovation the authors are claiming. The two boundary conditioning strategies—boundary forcing and coarse-resolution conditioning—appear to rely on interpolation or smoothing approaches that have been explored in prior regional downscaling and LAM studies. Could the authors clarify what is genuinely novel or insightful in these strategies or their findings?

2. Additionally, there seems to be an inconsistency between the text and figures: Section 5.1 states that four models (UNet, Stormer, GraphCast, Hi) were trained and evaluated, yet Figures 1 and 2 present results for six models (including FCN and Pangu-Weather). Could the authors clarify which models were actually trained and included in the benchmark results, and under what conditions?

---

### Official Review · Reviewer_yatZ · 2025-11-03

**Soundness:** 2
**Presentation:** 2
**Contribution:** 1
**Rating:** 4
**Confidence:** 5

**Summary:**

The paper presents a new dataset called IndiaWeatherBench to facilitate AI-based regional weather modeling over the Indian subcontinent/ The dataset is prepared using the IMDAA reanalysis dataset prepared from the Indian Meteorological Department. The dataset is claimed to be ML-ready and has been tested by training multiple neural network architectures ranging from UNet and Stormer to GraphCast and Hi (which is an improvement on GraphCast for hierarchical modeling).

Boundary conditions are important for regional modeling. Two experimental setups have been proposed for this:
(a) boundary forcing: where the padded high-resolution boundary conditions are provided using auxiliary variables
(b) coarse-grained weather forecasts: where the operational weather forecasts are downscaled to drive the forecasts by guiding them by providing more spatial information throughout the domain
While these ideas do not appear to be novel and have been applied elsewhere, they seem to be applied to the Indian context for the first time.

The performance of the models is tested using simple metrics like RMSE and the models have also been tested around the May 2019 heatwave.

**Strengths:**

The focus of the study on regional weather forecasting using AI is legitimate and requires more research. In this sense, the study addresses a topic that could be of importance to the weather forecasting community.

Though the novelty of the analysis is not clear to me (see comments below), the paper is written well, and the explanations are more or less clear, without the introduction of unnecessary jargon.

The experimental setup is sound. Though it could be further refined (see below), care has been taken to include multiple models and ML architectures/backbones in the analysis.

**Weaknesses:**

The study has multiple weaknesses:

1) Novelty: I fail to see a clear novelty in this work. It is not clear if the ML-ready dataset is the main outcome of the work, or if it is the freshly trained architectures or the imposition of boundary conditions.
---- Dataset: It is not clear if the IMDAA reanalysis is even better than the widely used ERA5 reanalysis. I would be surprised if it is better, but still, no effort has been made to even address any such differences
---- Models: By now, it is quite well-accepted that AIFS is the leading performing AI weather model. Multiple studies confirm this as well. So the state-of-the-art benchmark seems to be missing. If the novelty is the dataset, the inclusion of 4-5 different kinds of models does not make much sense to me, especially as the model performance is mixed
---- Boundary conditions: I appreciate the efforts to impose BCs in two different ways, but it seems that the idea itself is adopted from existing studies (pardon me if I am mistaken here). Moreover, since the experiment involves mixing multiple datasets, a big discussion on distribution shifts in the data is missing. You can't just mix and match datasets without any domain knowledge and hope that it magically works.

2) Mixed results with scant reasoning: Since the main motivation/goal of the work is not clear, I am further confused by the mixed performance from the models - the patterns in Figure 1, Figure 2, and Figure 4 are completely different. It does not help that the authors do not delve into any details on why the models behave the way they do. On the rare occasion that the authors hypothetize what might explain the difference in performance, I disagree with the reasoning presented. So, at best, I just read about a set of experiments with no clear goal or outcome.

3) No limitations addressed: The authors addressed a set of limitations. I agree with the mentioned limitations, but I also believe that some of those limitations must have been addressed in this work itself, rather than saving them up for later/future work, because they are fundamental to the study.
---- For instance, "(1) data" sure it can be generalised to other domains, but where will the data come from?
---- "(2) models" which ones did you miss here? This is a broad statement as the study includes many prevailing AI model architectures (with no clear winner architecture)
---- "(3) evaluation", this is the point I have issues with. The evaluation presented is shallow and based on the same old metrics like RMSE, which barely inform much about model performance and the spatial distribution of errors.

4) Originality: A quick Google search reveals that a similar evaluation of weather modeling over India exists (Gupta et al. (2025)), but has not been acknowledged in this study. I suggest the authors clarify in Section 2 how their study is different from this existing one, leading me to further question the original contributions of this work. Although Gupta wt al. don't seem to present a new dataset, the study arguably appears to provide a much more comprehensive evaluation of weather forecasting over the Indian subcontinent

References:
-- Lang, Simon, et al. "AIFS--ECMWF's data-driven forecasting system." arXiv preprint arXiv:2406.01465 (2024).
-- Gupta, Aman, Aditi Sheshadri, and Dhruv Suri. "MAUSAM: An Observations-focused assessment of Global AI Weather Prediction Models During the South Asian Monsoon." arXiv preprint arXiv:2509.01879 (2025).

**Questions:**

- I suggest Regional -> AI-based Regional in the title
- L28 - development in "AI-regional"
- L45-47: Please refrain from citing ALL the AI weather models out there. This is bad writing. Stick to 3-4 and add "for instance".
- Also, I wonder why Lang et al. (2024) is not mentioned here, since that seems to be the current state of the art anyway
- L49: Watson-Parris et al. (2022) is actually the Climate Bench paper and is inappropriately cited here. Rather, it is better to refer to it on L52 by adding ClimateBench.
- L73: What is the rationale behind using 6-hourly data apart from that WeatherBench uses 6-hourly data as well? Ideally, if you are creating a new dataset, you would want it to be at the highest resolution possible, especially for precipitation variables.
- Have you conducted a comparison of IMDAA vs ERA5? Regardless of a higher resolution, is IMDAA more accurate than ERA5? Is the distribution of chosen variables consistent with those in ERA5?
- Is the domain/spatial extent enough for weather prediction on 15 days?
- L261: mash -> mesh
- In all figures, why does PanguWeather show a sufficiently degraded performance even at 6-hour lead times? Something does not seem right here. I would not expect such short-term errors.
- Figure 1: What is the traditional baseline here? I only see ML errors. How do they compare to forecasts from (say) ECMWF or IMD forecasts? It could also be beneficial to add RMSE from AIFS here
- L370-371: prior efforts in weather foreasting ...: This is highly misleading. The prior efforts referred to were global prediction systems. This is a regional prediction system which is highly sensitive to boundary conditions. apples and oranges. Also if the novelty of the study is the dataset, then any such discussion seems tangential.
- L370-372: It would have been great if the authors provided some intuition as to why it doesn't work
- L419: Umm, I think PanguWeather is performing worse for most lead times and variables, isnt it? Not stormer. Please revise.
- L419-425: Can you please clarify if the lat-lon range of the bilinear interpolated coarse-resolution data is the same as the IMDAA reanalysis? If yes, then I do not think that the tokenization should introduce such errors. If the lat-lon range is difference, however, then I think the experimental setup is not correct and should account for identical lat-lon ranges.
- Figure 4: What is the source of the ground truth? IMDAA?
- Figure 4: Perhaps show anomaly wrt the ground truth instead? It is difficult to compare right now
- Note: It is impressive that Stormer gets the finer-scale features in the Southern parts/Western land boundaries of the sub-continent and no other models get it (not sure which state or district it corresponds to. Maybe the authors know it better)
- Section 5.2: In my opinion even Hi sort of underestimated the heat wave near the maximum regions in the north west. I would recommend analyzing/showing a histogram of temperature distributions over a chosen box instead.
- Figure 5: What is the source for the "reference"? I should emphasize that reanalysis (whether ERA5 or IMDAA) is NOT OBSERVATIONS.  So, it should not be treated as ground truth.

---

### Meta-Review · Area_Chair_3TTK · 2026-01-13

**Summary:**

The reviewer critiques are consistent and coherent. The core contribution remains ambiguous, failing to clearly establish novelty against existing datasets and models, including a notable omission of the AIFS state-of-the-art benchmark. Serious methodological concerns—particularly regarding the unanalyzed mixing of data sources and ambiguous validation protocols—undermine the technical soundness. Furthermore, the analysis is superficial, offering little interpretative insight into the mixed results, while critical limitations and relevant prior work are inadequately addressed.

**Reviewer Concerns:**

The major issues identified are, for the most part, fundamental and not easily corrected.

**Reviewer Scores:**

The reviewer critiques are consistent and coherent.

---

### Decision · Program_Chairs · 2026-01-26

Reject